# Large Language Models Can Self-Correct with Minimal Effort

**Zhenyu Wu** [1 2]   **Qingkai Zeng** [2]   **Zhihan Zhang** [2]   **Zhaoxuan Tan** [2]   **Chao Shen**[✉ 1]   **Meng Jiang** [2]

## Abstract

Intrinsic self-correct was a method that instructed large language models (LLMs) to verify and correct their responses without external feedback. Unfortunately, the study concluded that the LLMs could not self-correct reasoning yet. We find that a simple yet effective verification method can unleash inherent capabilities of the LLMs. That is to mask a key condition in the question, add the current response to construct a verification question, and predict the condition to verify the response. The condition can be an entity in an open-domain question or a numeric value in a math question, which requires minimal effort (via prompting) to identify. We propose an iterative verify-then-correct framework to progressively identify and correct (probably) false responses, named PROCO. We conduct experiments on three reasoning tasks. On average, PROCO, with GPT-3.5-Turbo as the backend LLM, yields +6.8 exact match on four open-domain question answering datasets, +14.1 accuracy on three arithmetic reasoning datasets, and +9.6 accuracy on a commonsense reasoning dataset, compared to Self-Correct.

## 1  Introduction

Reasoning is a cognitive process that uses evidence, arguments, and logic to arrive at conclusions or judgements (Huang & Chang, 2023). People have been exploiting and improving the reasoning ability of large language models (LLMs). Wei et al. proposed chain-of-thought (CoT) prompting and yielded promising results on several reasoning tasks, such as arithmetic reasoning (Kojima et al., 2022; Zhou et al., 2023), commonsense reasoning (Wei et al., 2022; Zhang et al., 2023; Wang et al., 2023b), and open-domain question answering (Wang et al., 2023a), using only

| Method | NQ | CSQA | AQuA |
|--------|------|------|------|
| CoT | 40.3 | 72.9 | 51.3 |
| Self-Correct | 40.1 | 65.9 | 48.7 |
| PROCO (Ours) | **48.0** | **75.5** | **65.2** |

Table 1: Performance comparison of different prompting methods using GPT-3.5-Turbo as backend LLM.

a few or no reasoning exemplars. CoT guides LLMs to generate intermediate reasoning steps instead of generating the final answer directly, which helps the LLMs simulate the human-like reasoning process.

Although CoT enables LLMs to handle some complex reasoning examples, it remains vulnerable to the negative impact of individual errors in each step. Specifically, even a minor error in one step can alter the trajectory of the entire reasoning process, ultimately leading to an incorrect conclusion. To address this issue, Dhuliawala et al.; Kim et al. have explored the verification and correction on the responses. For example, as shown in Figure 1**a**, for a given question and its initial LLM-generated answer, Self-Correct (Kim et al., 2023) first instructs the LLM to criticize its generated answer using the hint: "*Review previous answer and find mistakes*". Then, Self-Correct instructs the LLM to refine initial answers based on the critique.

However, recent studies (Huang et al., 2024; Gou et al., 2024) have cast doubt on the intrinsic self-correction capability of LLMs. Their research indicates that *without external feedback*, such as ground truth to verify the correctness of previous responses, LLMs struggle to correct their prior outputs. Since LLMs could not properly judge the correctness of their prior responses, the refined response might be even worse than the initial response.

To unleash inherent capabilities of LLMs to detect and rectify incorrect responses without external feedback, we introduce *substitute verification*. Let's look at a specific example. Given an open-domain question *"Who plays Skylar on Lab Rats: Elite Force?"*, we first prompt an LLM to generate an initial answer for the question, e.g., *"Paris Berelc"*. Next, we identify a key condition in the question that is relevant to the problem-solving process, such as *"Skylar"*. By masking the key condition in the question and adding the initial

[1]School of Cyber Science and Engineering, Xi'an Jiaotong University, China [2]Department of Computer Science and Engineering, University of Notre Dame, IN, USA. Correspondence to: Chao Shen <chaoshen@xjtu.edu.cn>.

*The first AI for MATH Workshop at the $41^{st}$ International Conference on Machine Learning*, Vienna, Austria. Copyright 2024 by the author(s).

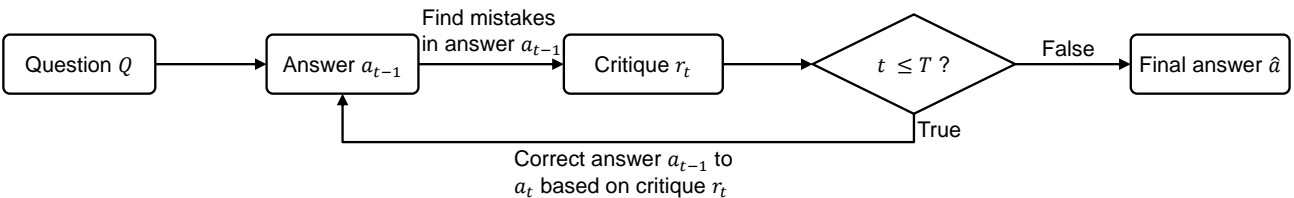

**a** Kim et al. proposed Self-Correct, instructing the LLM to critique and revise its answers using the hint "*Review previous answer and find mistakes.*" However, Huang et al. noted that LLMs struggle to correct mistakes without external feedback.

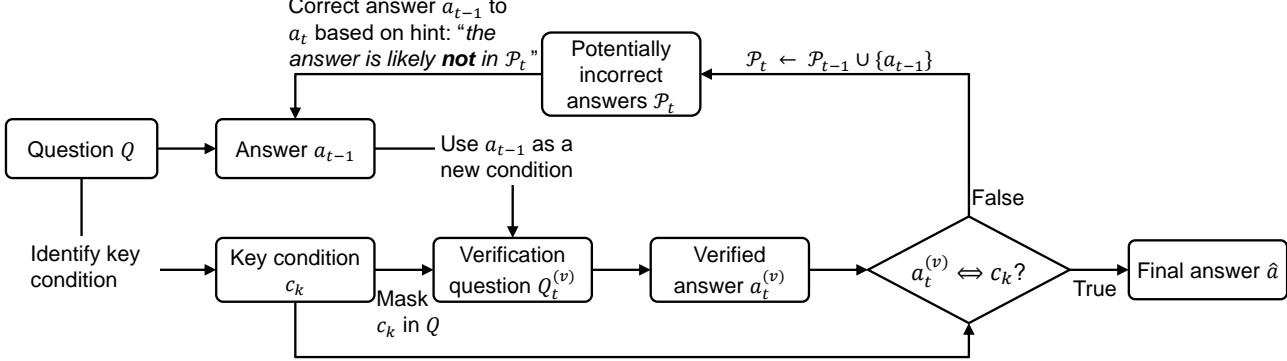

**b** PROCO performs three steps: (1) **Initialization**: Use CoT method to generate an initial answer. (2) **Verification**: Mask the key condition in the question and use the previous generated answer as a new condition to construct the verification question. Solve the verification question to get the verified answer and check if the verified answer and the key condition are equivalent. If they are equivalent, the previous generated answer is adopted as the final answer, otherwise add it to the set of potentially incorrect answers. (3) **Correction**: Use the set of potentially incorrect answers as feedback to correct previous generated answer. By cycle executing step (2) and step (3), the performance of LLMs on various complex reasoning tasks is progressively enhanced.

Figure 1: The proposed PROCO method helps LLMs identify incorrect answers and progressively correct them.

answer as a new condition, we can obtain a verification question: *"Who plays X on Lab Rats: Elite Force? Suppose the answer is Paris Berelc. What is the value of unknown variable X?"*. We use the LLM to solve the verification question, and we get that X is *"Skylar Storm"*. By verifying whether *"Skylar Storm"* is equivalent to *"Skylar"*, we can predict that the initial answer is likely correct.

Based on substitute verification, we propose a simple yet effective prompting method Progressive Correction (PROCO). Figure 1 illustrates the difference between the Self-Correct and PROCO methods. Compared with Self-Correct, our proposed PROCO, highlighting two primary distinctions:

**(1) Verification Method.** To improve verification accuracy, we propose the substitute verification method. PROCO first identifies key conditions relevant to the problem-solving process. It then masks one of the key conditions in the question and takes the generated answer as a new condition to construct the verification question. Finally, PROCO solves the verification question and gets the verified answer. If the verified answer and the key condition are equivalent, it indicates that the generated answer is likely to be correct.

**(2) Correction Method.** PROCO employs the substitute ver-

ification method to accurately verify the correctness of the LLM-generated answer. If an answer is deemed incorrect, PROCO adds it to the set of potentially incorrect answers, which then serves as feedback to guide LLMs in correcting previous mistakes with the hint: "*the answer is likely **not** in* {set of potentially incorrect answers}". By iteratively executing verification and correction, PROCO effectively prevents the repetition of previous mistakes, thereby progressively improving the quality of responses.

We conducted evaluations of PROCO using a variety of LLMs, including GPT-3.5-Turbo-1106, GPT-4-0125-Preview, and the open-source Mixtral-8x7B. These evaluations spanned three distinct tasks: arithmetic reasoning, commonsense reasoning, and open-domain question answering. The experimental results reveal that PROCO consistently outperforms existing methods. As shown in Table 1, PROCO achieves a 7.9 exact match (EM) improvement on the NQ dataset, a 16.5 absolute increase on the AQuA dataset, and a 9.6 absolute improvement on the CSQA dataset compared to the Self-Correct method.

In summary, our main contribution include:

- Based on our research, we have determined that LLMs

are capable of intrinsic self-correction, provided that the prompt design is carefully structured within a framework focused on verification and correctness.

- We introduce a novel prompting method, PROCO, which utilizes an iterative verify-then-correct framework. PROCO progressively refines responses by identifying key conditions and formulating verified questions specific to these conditions.

- We conduct extensive experiments across three different complex reasoning tasks and demonstrate that PROCO achieves significant improvements in both black-box and open-source LLMs.

## 2 Related Work

Self-Correct (Kim et al., 2023) methods, which aim to enhance the quality of LLM responses by providing feedback on initial attempts (Kim et al., 2023; Madaan et al., 2023; Chen et al., 2024), have demonstrated effectiveness in various reasoning tasks. These tasks include arithmetic reasoning (Madaan et al., 2023; Welleck et al., 2023), open-domain question answering (Dhuliawala et al., 2023; Yu et al., 2023b), commonsense reasoning (Kim et al., 2023), and others (Chen et al., 2024; Le et al., 2022). Self-Correct methods vary in the source and format of feedback, and the process of verifying the correctness of LLM output.

**Source and Format of Feedback** Interscript (Tandon et al., 2021) corrected the initial output of the LLM by integrating natural language feedback from humans. Due to the high cost of human feedback, various approaches have employed scalar reward functions as an alternative. For instance, Rainer (Liu et al., 2022) used reinforcement learning to generate contextual relevant knowledge in response to queries. Self-Correction (Welleck et al., 2023) trained a corrector to iteratively correct imperfect outputs. Other sources, such as compilers (Chen et al., 2024) or search engines (Yu et al., 2023b) can provide domain-specific feedback.

Recent research used LLMs to generate feedback. Self-Correct (Kim et al., 2023) and Self-Refine (Madaan et al., 2023) utilized LLMs to verify and refine their initial outputs. However, Huang et al. questioned the intrinsic self-correcting capability of LLMs, indicating that without external feedback, LLMs struggle to correct their previous responses. To unleash inherent capabilities of LLMs to detect and rectify incorrect responses without external feedback, we introduce *substitute verification*. By providing natural language feedback based on verification results, we can steer LLMs away from incorrect answers, thus enhancing their performance in various reasoning tasks.

**Verify Correctness of LLM Output** Several studies trained or fine-tuned language models to check the correct-

Figure 2: Key conditions in complex reasoning tasks play a crucial role in the problem-solving process. These conditions can take various forms: a numeric value in arithmetic questions, an entity in open-domain questions, or a concept in commonsense questions.

ness of answers. Cobbe et al. fine-tuned GPT-3 as a verifier to judge the correctness of solutions. Li et al. fine-tuned DeBERTa-v3-large (He et al., 2021) to predict the probability that the generated reasoning path leads to a correct answer. Lightman et al. constructed a large dataset with step-wise correctness labels from human annotators, and fine-tuned a GPT-4 model on it. These methods require significant human annotations. To reduce human labor, Peng et al. proposed using an external database to identify incorrect knowledge in LLM outputs. Chern et al. used tools for fact-checking. Miao et al. used the LLM to verify the correctness of each step in the arithmetic reasoning path based on preceding steps. Dhuliawala et al. used manually crafted demonstrations as context to prompt the LLM to check the correctness of its output. All of these methods solely verify the correctness of LLM outputs and select the verified answer as the final answer. In contrast, our method iterates a verify-then-correct process to progressively identify and rectify incorrect answers.

## 3 Preliminaries

Given a question $Q$, consisting of $m$ context sentences $\{s_j\}_{j=1}^m$ and one query sentence $q$. The query $q$ ends with a question mark and is usually the last sentence of $Q$. We can express $Q = (\oplus_j s_j) \oplus q$, where $\oplus$ denotes text concatenation function. We extract conditions $\{c_i\}_{i=1}^n$ that are numerical values (arithmetic reasoning), entities (open-domain question answering), and concepts (commonsense reasoning), as shown in Figure 2. It is worth noting that usually $n \geq m$, if the question has one or multiple conditions. We denote $J(i) \in \{1, \ldots, m\}$ as the index of the context sentence containing the condition $c_i$. Among these conditions, the key condition $c_k$ is crucial for problem-solving and is used in the substitute verification process, where $k$ is the index of the key condition within $\{c_i\}_{i=1}^n$. We introduce two innovative approaches for identifying the key condition.

**Similarity-based Key Condition Identification**  Numerical values are crucial in arithmetic reasoning tasks, so we select those relevant to solving the problem as key conditions. Key conditions are found in context sentences $\{s_j\}_{j=1}^m$ with high semantic relevance to the query sentence $q$. We use SimCSE (Gao et al., 2021) to encode the context and the query sentences, represented as $\{\mathbf{s}_j\}_{j=1}^m$ and $\mathbf{q}$, respectively. Semantic relevance is calculated using cosine similarity between $\{\mathbf{s}_j\}_{j=1}^m$ and $\mathbf{q}$. The most relevant context sentence index $\ell$ is determined by:

$$\ell = \operatorname{argmax}_{j \in \{1,\ldots,m\}} \cos(\mathbf{s}_j, \mathbf{q}). \tag{1}$$

We use regular expressions to extract the numerical value in context sentence $s_\ell$ as the key condition $c_k$. If multiple numerical values are present, one is randomly selected as the key condition.

**Zero-shot Key Condition Identification**  Identifying key conditions in open-domain question answering (Entity) and commonsense reasoning (Concept) is not possible through regular expressions, unlike in arithmetic reasoning (Numerical Value). Instead, we directly instruct LLMs to identify these relevant entities or concepts as key conditions. For instance, given an open-domain question $Q$, we construct a key condition identification prompt: "*Given the question below, the task is to identify a set of entities within the question and then select the one that is most relevant to the problem-solving process. Q*". We then input this prompt into an LLM to obtain the key condition $c_k$.

# 4  Proposed Approach

## 4.1  Overview

In this section, we present the overall pipeline of the proposed Progressive Correction (PROCO) prompting method which consists of three steps. Figure 1**b** illustrates the PROCO method. Initially, PROCO prompts the LLM to generate an answer in response to a given question (Sec. 4.2). Subsequently, to enhance the preliminary answer, PROCO identifies a key condition and generates a corresponding verification question-answer pair based on that condition (Sec. 4.3). The final answer is refined by verifying the question-answer pair, ensuring the answer's consistency and accuracy (Sec. 4.4). The full prompts used in the experiments can be found in Appendix A.2.

## 4.2  Generate Initial Answer

Given a question $Q$, we use one of the existing prompting methods, such as CoT (Kojima et al., 2022), RAG (Khattab et al., 2023), or GenRead (Yu et al., 2023a), to generate an initial answer $a_0$. By default, we use the CoT (Kojima et al., 2022) prompting method to generate an initial answer.

## 4.3  Iterative Verify-then-Correct Process

We propose a novel iterative verify-then-correct method that first initializes the set of potentially incorrect answers as an empty set $\mathcal{P}_0 = \varnothing$ and identifies the key condition $c_k$ within the question $Q$ (Sec. 3). The method then progressively corrects the LLM-generated answer over $T$ iterations by cyclically conducting verification and correction phases. Here we use the $t$-th iteration as an example to illustrate the verify-then-correct process.

**Verification Phase**  The verification phase uses substitute verification method to verify the correctness of the previous generated answer $a_{t-1}$. It includes several substeps.

First, the key condition $c_k$ within the question $Q$ is replaced with a specific token "X" , resulting in a mask question:

$$Q^{(\text{mask})} = \left( \oplus_j s_j \big|_{s_{J(k)} = s_{J(k)}^{(\text{mask})}} \right) \oplus q. \tag{2}$$

where $s_{J(k)}$ is the context sentence containing the key condition $c_k$, $s_{J(k)}^{(\text{mask})}$ denotes replacing $c_k$ in $s_{J(k)}$ with "X". We then construct the $t$-th verification question $Q_t^{(v)}$ based on the mask question:

$$Q_t^{(v)} = Q^{(\text{mask})} \oplus a_{t-1} \oplus q^{(v)} \tag{3}$$

where $q^{(v)}$ is a static question for verification, e.g., "*What is the value of the unknown variable X?*" Note that through all iterations, the key condition remains the same, and we do not use it to construct $Q_t^{(v)}$, for any $t \in \{1, \ldots, T\}$. The LLM is instructed to solve the verification question $Q_t^{(v)}$ and produce the corresponding answer $a_t^{(v)}$. Finally, different strategies are proposed to verify the correctness of $a_{t-1}$.

**Match-based Verification.** For arithmetic questions, if $a_t^{(v)}$ is equal to $c_k$, it indicates that the previous answer $a_{t-1}$ is most likely correct.

**Proposition-based Verification.** For open-domain or commonsense questions, we propose a proposition-based verification method to verify the correctness of the previously generated answer $a_{t-1}$. The intuition behind this is that the question $Q_t^{(v)}$ may have multiple valid answers, and directly checking if $a_t^{(v)}$ exactly matches $c_k$ could result in misclassifying a correct answer as incorrect. Specifically, we construct an answer verification prompt: "*Determine the correctness of the proposition: If the answer to question $Q_t^{(v)}$ is $c_k$, then X could also be $a_t^{(v)}$*". We input this prompt into an LLM and receive a judgment about the proposition's correctness. If the proposition is verified as correct, it indicates that the previously generated answer $a_{t-1}$ is likely correct, and we select $a_{t-1}$ as the final answer $\hat{a}$ and exit the loop. Otherwise, we add $a_{t-1}$ to the set of potentially incorrect answers $\mathcal{P}_{t-1}$ to obtain the updated set $\mathcal{P}_t$.

**Correction Phase** During the correction phase, we use the set of potentially incorrect answers $\mathcal{P}_t = \{a_0, \cdots, a_{t-1}\}$ as feedback to generate a corrected answer $a_t$. For a given question $Q$ and the set $\mathcal{P}_t$, we append the phrase "*the answer is likely not in $\mathcal{P}_t$*" to the question. This instructs the large language model to re-answer the question while avoiding repeating previous mistakes.

### 4.4 Final Answer Determination

The process of verify-then-correct can be iterated until specific stopping conditions are met. This process terminates under three situations: First, if the answer $a_{t-1}$ is verified to be likely correct, it is selected as the final answer. Second, if the corrected answer $a_t$ matches the previously generated answer $a_{t-1}$, then $a_t$ is chosen as the final answer. Lastly, if the iteration count surpasses the maximum number of iterations $T$, the last LLM-generated answer $a_T$ is adopted as the final answer.

## 5 Experiments

### 5.1 Experimental Setup

**Datasets.** We evaluate PROCO on three complex reasoning tasks: arithmetic reasoning (GSM8K (Cobbe et al., 2021b), AQuA (Ling et al., 2017), and MATH (Hendrycks et al., 2021)); open-domain question answering (NQ (Kwiatkowski et al., 2019), TriviaQA (Joshi et al., 2017), WebQ (Berant et al., 2013), and HotpotQA (Yang et al., 2018)); and commonsense reasoning (CSQA (Talmor et al., 2019)). Detailed information about these datasets is available in Appendix A.1.

**Baselines.** To verify the effectiveness of our method, we compare PROCO with three principal baseline categories: Using LLMs to generate problem-related documents: GenRead (Yu et al., 2023a) prompts an LLM to generate $M$ contextual documents based on a given question and then reads these documents to produce the final answer. Using search engines to retrieve problem-related documents: RAG (Khattab et al., 2023) retrieves $M$ relevant documents from Bing search[1] based on a given question and then prompts an LLM to read the retrieved documents to produce the final answer. Direct question answering without external documents: CoT (Kojima et al., 2022) appends "*Let's think step by step*" to the given question, instructing the LLM to generate a reasoning path leading to the final answer. CoVe (Dhuliawala et al., 2023) answers the given question, generates verification questions, answers them, and produces the final answer based on the verification results. Self-Correct (Kim et al., 2023) instructs an LLM to critique and refine its initial response. We use all methods as baselines for open-

domain question answering and commonsense reasoning tasks. For arithmetic reasoning, where external documents are unnecessary, CoT and Self-Correct serve as baselines. These baseline methods can be integrated into PROCO. For example, we can use the GenRead (Yu et al., 2023a) method to generate an initial answer for a given question and use our proposed PROCO method to progressively correct the initial answer (i.e., GenRead + PROCO).

**Evaluation Metrics.** In open-domain question answering, we use exact match (EM) score and F1 score to evaluate model performance (Zhu et al., 2021). For the EM score, an answer is considered correct if and only if its normalized form (Yu et al., 2023a) has a match in the acceptable answer list. The F1 score treats the prediction and ground truth as bags of tokens, and computes the average overlap between them. For other complex reasoning tasks, we use accuracy as the evaluation metric.

**Implementation.** We evaluate PROCO across three LLMs of different scales: GPT-3.5-Turbo-1106 and GPT-4-0125-Preview, which are the most widely used LLMs with public available APIs[2]. Additionally, we include Mixtral-8x7B[3] (Jiang et al., 2024), an open source LLM with 47 billion parameters. For baselines like GenRead (Yu et al., 2023a) and RAG (Khattab et al., 2023) that use external documents, we set the number of documents $M = 5$. When incorporating these methods with PROCO, we set $M = 1$. The temperature parameter is set to $0.7$ in our experiments.

### 5.2 Experimental Results

**Overall performance on open-domain question answering and commonsense reasoning tasks.** Table 2 demonstrates that PROCO significantly enhances problem-solving performance across five benchmarks when combined with baseline methods using external documents. This improvement holds for both black-box and open-source LLM backends. Specifically, for GPT-3.5-Turbo-1106, using GenRead to generate an initial answer and then correcting it with PROCO (GenRead + PROCO) boosts the EM score by +6.1 on NQ, +7.6 on TriviaQA, +5.4 on WebQ, +9.0 on HotpotQA, and improves accuracy by +9.1 on CSQA.

Without external documents, PROCO shows superior self-correctness compared to Self-Correct and CoVe. It achieves gains of +7.9, +7.4, +7.8, +4.0, and +9.6 on NQ, TriviaQA, WebQ, HotpotQA, and CSQA, respectively, compared to Self-Correct. Additional experimental results are shown in Appendix A.3.

**Overall performance on arithmetic reasoning tasks.** For arithmetic reasoning tasks, we compare PROCO only

---

[1] https://www.microsoft.com/en-us/bing/apis/

[2] https://platform.openai.com/docs/models
[3] https://github.com/mistralai/mistral-src

| Method | Open-domain Question Answering | | | | | | | | Commonsense Reasoning |
|---|---|---|---|---|---|---|---|---|---|
| | NQ | | TriviaQA | | WebQ | | HotpotQA | | CSQA |
| | EM | F1 | EM | F1 | EM | F1 | EM | F1 | Accuracy |
| *Using LLMs to generate problem-related documents* | | | | | | | | | |
| GenRead | 42.2 / 46.7 | 49.4 / 52.0 | 70.8 / 69.0 | 74.8 / 72.4 | 41.3 / 51.1 | 48.5 / 56.5 | 38.0 / 36.0 | 43.2 / 39.7 | 67.3 / 64.3 |
| GenRead + PRoCo | 48.3 / 48.5 | 55.6 / 53.7 | 78.4 / 72.3 | **82.4** / 75.8 | 46.7 / 52.0 | 53.9 / 57.5 | **47.0** / 38.0 | **51.0** / 42.3 | **76.4** / 70.4 |
| *Using search engines to retrieve problem-related documents* | | | | | | | | | |
| RAG | 45.3 / 48.8 | 52.4 / 54.6 | 72.7 / 75.3 | 76.4 / 78.5 | 40.1 / 46.3 | 46.9 / 52.1 | 37.0 / 37.0 | 41.1 / 40.2 | 65.9 / 66.3 |
| RAG + PRoCo | **48.5 / 51.6** | **56.0 / 57.1** | 78.4 / **79.6** | 82.1 / **83.0** | 45.2 / 50.3 | 52.5 / 56.3 | 39.0 / **41.0** | 44.2 / **43.7** | 74.2 / 71.8 |
| *Direct question answering without external documents* | | | | | | | | | |
| CoT | 40.3 / 42.6 | 46.4 / 48.2 | 69.2 / 66.7 | 72.2 / 70.3 | 38.2 / 46.6 | 44.6 / 51.9 | 28.0 / 29.0 | 31.2 / 34.4 | 72.9 / 68.4 |
| Self-Correct | 40.1 / 44.8 | 47.1 / 50.5 | 71.3 / 71.3 | 74.1 / 74.8 | 39.2 / 47.5 | 45.7 / 51.9 | 29.0 / 32.0 | 32.4 / 36.2 | 65.9 / 49.8 |
| CoVe | 43.4 / 47.6 | 48.9 / 53.0 | 76.4 / 73.2 | 79.4 / 76.4 | 43.1 / 53.4 | 49.0 / 58.2 | 31.0 / 33.0 | 35.2 / 36.9 | 73.1 / 70.8 |
| PRoCo | 48.0 / 50.7 | 54.8 / 53.6 | **78.7** / 74.5 | 82.1 / 76.6 | **47.0 / 55.1** | **57.0 / 59.2** | 33.0 / 35.0 | 36.2 / 41.3 | 75.5 / **72.7** |

Table 2: Performance on NQ, TriviaQA, WebQ, HotpotQA, and CSQA benchmarks using GPT-3.5-Turbo-1106 (black-box LLM) and Mixtral-8x7B (open-source LLM). Each cell shows GPT-3.5-Turbo-1106 / Mixtral-8x7B performance. The best performance for each dataset is highlighted in bold. PRoCo improves baseline methods with external documents across all benchmarks and outperforms those without external documents.

| Method | Arithmetic Reasoning | | |
|---|---|---|---|
| | GSM8K | AQuA | MATH |
| CoT | 78.6 / 74.4 | 51.3 / 49.2 | 37.9 / 28.4 |
| Self-Correct | 75.1 / 72.5 | 48.7 / 44.4 | 27.6 / 21.5 |
| PRoCo | **87.1 / 78.7** | **65.2 / 54.3** | **41.5 / 30.2** |

Table 3: Accuracy on arithmetic reasoning tasks. Each cell shows GPT-3.5-Turbo-1106 / Mixtral-8x7B performance.

| Method | GSM8K | CSQA | HotpotQA |
|---|---|---|---|
| | Accuracy | Accuracy | EM |
| CoT | 95.5 | 82.0 | 49.0 |
| Self-Correct | 91.5 | 79.5 | 49.0 |
| CoVe | - | 83.5 | 57.0 |
| PRoCo | **97.6** | **86.7** | **61.0** |

Table 4: Performance comparison of baseline methods using GPT-4-0125-Preview on three types of reasoning tasks: accuracy in GSM8K and CSQA, and EM score in HotpotQA.

| Method | NQ | | TriviaQA | | WebQ | |
|---|---|---|---|---|---|---|
| | EM | Tokens | EM | Tokens | EM | Tokens |
| GenRead | 42.2 | 1023.3 | 70.8 | 924.2 | 41.3 | 963.3 |
| GenRead + PRoCo | 48.3 | 469.1 | 78.4 | 465.0 | 46.7 | 416.8 |
| Δ | 14.5% ↑ | 54.2% ↓ | 10.7% ↑ | 49.7% ↓ | 13.1% ↑ | 56.7% ↓ |
| RAG | 45.3 | 1971.5 | 72.7 | 1937.5 | 40.1 | 2067.8 |
| RAG + PRoCo | 48.5 | 916.4 | 78.4 | 968.2 | 45.2 | 875.5 |
| Δ | 7.1% ↑ | 53.5% ↓ | 7.8% ↑ | 50.0% ↓ | 12.7% ↑ | 57.7% ↓ |

Table 5: Comparison of PRoCo with baselines including external documents: Efficiency and Effectiveness. PRoCo consistently outperforms baselines on all benchmarks using significantly fewer tokens.

with CoT and Self-Correct, as baselines with external documents and CoVe are unsuitable. As shown in Table 3, PRoCo demonstrates superior self-correctness over all baseline methods across benchmarks on both black-box and open-source LLMs. Specifically, when applied to GPT-3.5-Turbo-1106, PRoCo improves accuracy by an average of 14.1 compared to the Self-Correct.

**PRoCo with GPT-4 as backbone model.** We compare PRoCo with baseline methods using the GPT-4-0125-Preview model to test its effectiveness. Due to the high

cost of GPT-4-0125-Preview, we select GSM8K for arithmetic reasoning, HotpotQA for open-domain question answering, and CSQA for commonsense reasoning. Only baseline methods without external documents are included. As shown in Table 4, PRoCo outperforms the baselines across all benchmarks with the GPT-4 model.

**Retrieve External Documents vs. PRoCo.** Since both retrieve external documents (RAG / GenRead) and verify-and-correct (PRoCo) can enhance the performance on complex reasoning via adding the token cost for each question, we discuss the trade-off between efficiency and effectiveness to apply them to real-world reasoning task. Table 5 shows PRoCo outperforms GenRead/RAG in EM scores across three open-domain question-answering benchmarks, using just one external document compared to five. PRoCo achieves an average 12.8% higher EM score than GenRead and 9.2% higher than RAG, while using half the tokens. Further analysis shows that multiple external documents

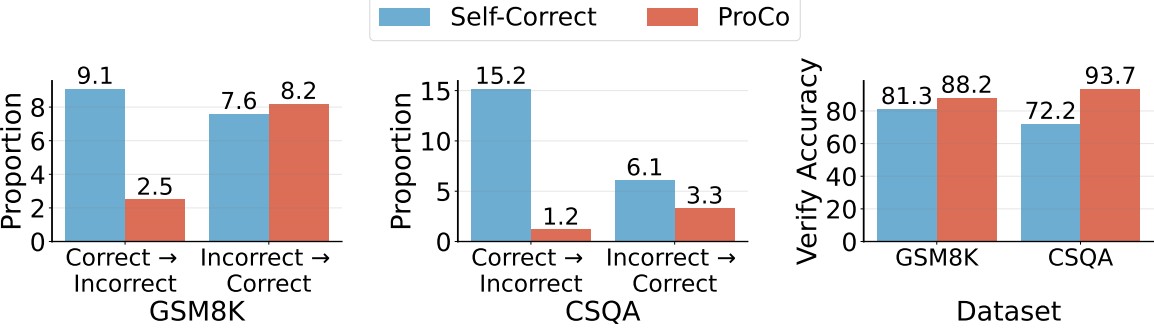

Figure 3: Analysis of changes in answers after three rounds of corrections. Correct → Incorrect: A correct answer is changed to an incorrect one. Incorrect → Correct: An incorrect answer is revised to a correct one. Self-Correct is more likely to modify correct answers to an incorrect ones, rather than fixing erroneous predictions. PROCO can properly judge the correctness of the answers, and correct wrong answers.

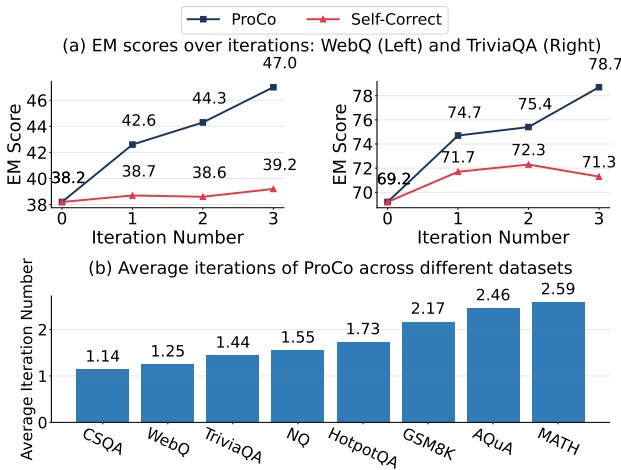

Figure 4: Analysis of Iterations in PROCO: PROCO shows improved performance with an increased number of iterations. Across all benchmarks, a minimum of three iterations ensures PROCO has ample opportunity to verify and correct answers generated by LLMs.

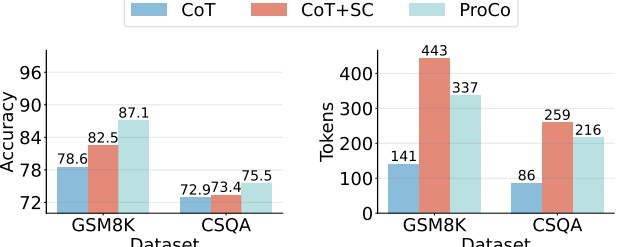

Figure 5: Performance comparison of CoT, PROCO, and CoT with self-consistency (i.e., CoT + SC). Compared to CoT + SC, PROCO not only exhibits higher accuracy, but also consumes fewer tokens.

often contain excessive irrelevant or redundant information, leading to incorrect answers and unnecessary token costs.

## 5.3 Analysis in PROCO

**Analysis of Self-Correctness in PROCO** Figure 3 shows the impact of PROCO after three correction rounds using GPT-3.5-Turbo-1106. PROCO is more accurate than Self-Correct in identifying errors in LLM-generated answers, with an 6.9% improvement in GSM8K and a 21.5% improvement in CSQA. Furthermore, PROCO is more effective at correcting errors without introducing new ones, enhancing LLM performance in complex reasoning tasks. For GSM8K, PROCO incorrectly alters correct answers 2.5% of the time and corrects incorrect answers 8.2% of the time, compared

to Self-Correct, which makes correct answers incorrect 9.1% of the time and fixes incorrect answers 7.6% of the time.

**Impact of Iteration Count.** Figure 4 (a) shows that the EM score of PROCO improves with more iterations in both WebQ and TriviaQA. In contrast, Self-Correct shows minimal improvement in WebQ and even a 1.0 EM score decrease in TriviaQA with more iterations. Additionally, the average iteration number for PROCO across all benchmarks is less than 3. Considering the trade-off between efficiency and effectiveness, we set the maximum number of iterations to 3 per question.

**Comparison between Self-consistency and PROCO** Since PROCO iteratively corrects answers for complex reasoning tasks, we propose that Self-consistency (SC) (Wang et al., 2023c), which solves a problem multiple times and uses a majority vote to determine the final answer, may reduce errors by minimizing bias and enhancing the robustness of LLM performance.

We evaluate the performance of CoT with self-consistency

| Question | Where does patience is a virtue come from? |
|---|---|
| **Gold Answer** | Psychomachia |
| **CoT** | The phrase "patience is a virtue" is an old proverb that has been passed down through generations. Its exact origin is not known, but it is believed to have originated from ==ancient wisdom and moral teachings==. ✗ |
| **Self-Correct** | The phrase "patience is a virtue" comes from the English poet William Langland's work ==Piers Plowman== in the 14th century. ✗ |
| **CoVe** | "Patience is a virtue" is a well-known proverb that has been in use since at least the 14th century. It is often attributed to the Latin writer ==Publilius Syrus==. ✗ |
| **PROCO** | The phrase "patience is a virtue" can be traced back to medieval European literature. For instance, in the ==Psychomachia==, a 5th-century poem by Prudentius, patience is depicted as one of the virtues. ✓ |
| **RAG** | The first known recording of the expression in English is in the poem ==Piers Plowman== by William Langland, written around the year 1360. Langland's poem includes the line "Patience is a high virtue". (`https://www.bookbrowse.com/expressions/detail/index.cfm/expression_number/416/patience-is-a-virtue`) ✗ |
| **RAG + PROCO** | The idea behind the expression "patience is a virtue" dates back to the 5th century. It can be found in the epic poem ==Psychomachia==. (`https://writingexplained.org/idiom-dictionary/patience-is-a-virtue`) ✓ |

Table 6: Case study of answers generated by various methods. The final answer is highlighted in yellow. PROCO shows superior self-correction compared to baseline methods that include self-correction processes. Additionally, PROCO reduces errors generated by methods that use external documents, ensuring correct source citation.

| Method | TriviaQA | CSQA |
|---|---|---|
| Equivalent | **82.4** | **93.7** |
| Match | 40.2 | 29.7 |
| Similarity | 69.2 | 65.9 |

Table 7: Accuracy of different verification methods. "Equivalent" determines the correctness of an LLM-generated answer by checking whether the answer to the verification question and the key condition are equivalent. "Match" determines the correctness of an LLM-generated answer by checking whether the answer to the verification question exactly matches the key condition. "Similarity" indicates that the correctness of an LLM-generated answer is determined by evaluating the semantic similarity between the answer to the verification question and the key condition.

(CoT + SC) on two complex reasoning tasks (GSM8K and CSQA) and compare it with PROCO. For a fair comparison, CoT + SC generates answers three times per question, matching ProCo's maximum iterations. As shown in Figure 5, we find that PROCO uses fewer tokens and achieves better accuracy on both tasks. This is because, unlike PROCO's verification and correctness processes, CoT + SC merely solves the problem multiple times, often repeating the same mistakes.

**Can we just use the exact match method during the verification phase?** In PROCO, we consider an LLM-generated answer to be correct as the answer to the verification ques-

tion and the key condition are equivalent; we denote this method as "Equivalent". To evaluate our answer verification approach, we also consider: (1) "Match", where an LLM-generated answer is deemed correct if it exactly matches the key condition, and (2) "Similarity", where an LLM-generated answer is considered correct if it is semantically similar to the key condition. As shown in Table 7, verifying whether the answer to the verification question is equivalent to the key condition can accurately assess the correctness of LLM-generated answers.

### 5.4 Case Study

Table 6 shows that, except for RAG + PROCO and PROCO, all other methods fail to provide the correct answer to the given problem. CoT generates an incorrect answer, unable to determine the origin of the phrase "Patience is a virtue". Self-Correct, CoVe, and RAG erroneously assert that the phrase originated in the 14th century. In contrast, RAG + PROCO and PROCO accurately identify the first appearance of the phrase "Patience is a virtue" in the 5th century. Furthermore, RAG + PROCO provides the correct source for citation. This indicates that integrating RAG into PROCO significantly enhances answer accuracy and reliability.

## 6 Comparison Between Self-verification and PROCO

Compared to Self-verification (Weng et al., 2023), PROCO exhibits several key distinctions:

- **Research Background:** Self-verification addresses the limitations of existing methods that rely on manually annotated data to train verifiers for scoring LLM-generated responses. It introduces LLMs as verifiers to improve the selection of better prediction results. In contrast, PROCO is inspired by observations from the results of Self-Correct (Huang et al., 2024), not Self-verification. It focuses on developing a better "correction" (not verification) solution for general reasoning tasks, utilizing almost all datasets tested in Self-Correct. PROCO makes a significant contribution to correct reasoning responses by LLM themselves, compared to the conclusion from Self-Correct. The introduction section of PROCO clearly outlines its motivations and contributions.

- **Problem Identification:** Self-verification primarily aims to enhance the performance of LLMs in solving math word problems by selecting the highest-scoring response from multiple LLM-generated answers. Its primary contribution lies in answer verification. While Self-verification has also been tested on commonsense question answering, the observed improvements are minimal, likely due to the suboptimal definition of conditions or the lack of effective methods for condition extraction, key condition identification, and rectification in general reasoning tasks. In contrast, PROCO is not specifically focused on mathematical reasoning. Instead, it aligns closely with correction-based methods, which are extensively cited in PROCO. PROCO introduces the use of "key" concepts and "key" entities as conditions. Identifying these "key" conditions is a non-trivial task, and verifying them while leveraging the verification results for iterative correction presents additional challenges. This aspect highlights the solid technical contributions of PROCO.

- **Method Implementation:** First, the frameworks of Self-verification and PROCO differ significantly. Self-verification aims to verify the correctness of LLM-generated answers. It looks for a score. If LLM generates multiple answers, the scores may find the best one among them. In contrast, PROCO aims to construct a set of "potentially incorrect answers" so the LLM can be aware of it and rectify or correct its original answers. Unlike Self-verification, which relies on scoring, PROCO emphasizes iterative feedback to enhance answer accuracy through correction. Second, the verification modules of Self-verification and PROCO differ significantly. Self-verification randomly selects a numerical value from the question and replaces its occurrence with a special token X. We considered Self-verification as a related work for PRP (Wu et al., 2024), because both works tried to use X to replace numeric values in a question. In contrast, PROCO identifies key conditions critical to the problem-solving process,

masks one of these conditions in the question, and takes the generated answer as a new condition to formulate the verification question. These key conditions are essential for complex reasoning tasks and can take various forms, such as numerical values in arithmetic questions, entities in open-domain questions, or concepts in commonsense questions. This method is broadly applicable to diverse reasoning tasks. Third, the final answer determination modules of Self-verification and PROCO differ significantly. Self-verification selects the highest-scoring response from multiple LLM-generated answers as a result. In contrast, PROCO selects the answer which is verified to be likely correct as the final answer. A key distinction lies in their handling of errors: Self-verification fails when the correct answer is not included in the sample set, whereas PROCO utilizes an iterative verify-and-correct framework that mitigates the repetition of mistakes, thereby improving LLM performance in complex reasoning tasks.

- **Examples Provided:** This paper illustrates PROCO's substitute verification process using a case from open-domain question answering, which involves identifying key conditions, constructing verification questions, and applying a proposition-based verification method. In contrast, Self-verification employs a case from GSM8K to highlight the details of its backward verification process. The two papers differ not only in the examples used but also in the verification processes we aim to clarify.

- **Prompt Details:** PROCO employs a specially crafted prompt that emphasizes verification and iterative correction. Unlike the scoring-based prompts used in Self-verification, PROCO's prompt explicitly instructs the identification and rectification of incorrect answers. Its structure includes the following steps: zero-shot initial answer generation, key condition identification, equivalence checking, and correction of errors. In contrast, Self-verification uses a few-shot approach for both initial answer generation and backward verification. These differences underscore the distinct prompt structures and the fundamentally different roles they play in their respective methodologies.

## 7 Conclusion

In this study, we present a novel zero-shot prompting method for solving complex reasoning tasks. We name it progressive correction (PROCO), which first prompts a large language model to generate an initial response, then iterates a verify-then-correct process to progressively identify and correct (probably) false responses. Extensive experiments on eight complex reasoning datasets demonstrate the effectiveness and efficiency of our proposed method.

## Limitations

This study focused solely on complex reasoning tasks in English, excluding non-English tasks from our training and test data. Consequently, the method may not perform well for non-English tasks. Future research will explore solutions for multilingual complex reasoning tasks.

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

# A Appendix

## A.1 Datasets

We evaluate PROCO on three complex reasoning tasks: arithmetic reasoning (GSM8K (Cobbe et al., 2021b), AQuA (Ling et al., 2017), and MATH (Hendrycks et al., 2021)); open-domain question answering (NQ (Kwiatkowski et al., 2019), TriviaQA (Joshi et al., 2017), WebQ (Berant et al., 2013), and HotpotQA (Yang et al., 2018)); and commonsense reasoning (CSQA (Talmor et al., 2019)). All of these datasets are accessible under the MIT License. Below, we provide brief descriptions of the datasets used:

- GSM8K (Cobbe et al., 2021b) consists of high quality grade school math word problems created by human problem writers. These problems require 2 to 8 steps to solve, and solutions primarily involve performing a sequence of elementary calculations using basic arithmetic operations to reach the final answer.
- AQuA (Ling et al., 2017) contains multiple-choice math questions that cover a broad range of topics and difficulty levels.
- MATH (Hendrycks et al., 2021) is a challenging datasets consisting of 12k problems across seven categories, testing models' advanced math and science reasoning. The problems in this dataset are very hard as they come from mathematics competitions written in LaTeX.
- NQ (Kwiatkowski et al., 2019) were collected from real Google search queries and the answers are one or multiple spans in Wikipedia articles identified by human annotators.
- TriviaQA (Joshi et al., 2017) includes trivia questions with answers originally scraped from trivia and quiz-league websites.
- WebQ (Berant et al., 2013) consists of questions selected using Google Suggest API, where the answers are entities in Freebase.
- HotpotQA (Yang et al., 2018) contains 113k multi-hop questions in natural language. The questions are collected by crowdsourcing based on Wikipedia articles with human annotated supporting evidence and answers.
- CSQA (Talmor et al., 2019) offers a collection of multiple-choice questions testing commonsense reasoning. We use the development set for our evaluation.

## A.2 Full Prompts in Experiments

### A.2.1 ARITHMETIC REASONING

Given an arithmetic question $Q$, we use the CoT prompting method to generate an initial answer. Specifically, we first construct a reasoning generation prompt: "Q: $Q$. A: Let's think step by step." as shown in Prompt A.1. We then

feed the above prompt to the LLM, which subsequently generates a reasoning path. To extract the answer from the reasoning path, we append an answer extraction instruction, creating the numerical answer extraction prompt: "Q: $Q$. A: {reasoning path} The answer (arabic numerals) is:" as shown in Prompt A.2.

---

**Prompt A.1: Initial Answer Generation**

Q: $Q$
A: Let's think step by step.

---

**Prompt A.2: Numerical Answer Extraction**

Q: $Q$
A: {reasoning path} The answer (arabic numerals) is:

---

We use the substitute verification method to verify the correctness of the previous generated answer. Specifically, we first identify the key condition within the question (Sec. 3). By replacing the key condition with a specific token X, we create a masked question. We then append the sentence, "Suppose the answer is {previous generated answer}. What is the value of unknown variable X?" to the masked question to formulate the verification question, as shown in Prompt A.3.

---

**Prompt A.3: Verification Question Construction**

{masked question} Suppose the answer is {previous generated answer}. What is the value of unknown variable X?

---

Using Prompt A.1 and Prompt A.2, we can obtain the numerical answer for the verification question. By checking if the numerical answer for the verification question is equal to the key condition, we can assess the correctness of the previous generated answer. If the previous generated answer is deemed incorrect, we add it to the set of potentially incorrect answers; otherwise, we select it as the final answer. For incorrect answers, we use the Prompt A.4 to correct them.

---

**Prompt A.4: Incorrect Answers Correction**

Q: $Q$ (the answer is likely not in {set of potentially incorrect answers})
A: Let's think step by step.

---

### A.2.2  OPEN-DOMAIN QUESTION ANSWERING

Given an open-domain question $Q$, we use the Prompt A.2 to instruct the LLM to generate a reasoning path. To extract the answer from this reasoning path, we add an answer extraction instruction, resulting in the following entity answer extraction prompt: "Answer the following question with just one entity. Q: $Q$. A: {reasoning path} The answer is:" as shown in Prompt A.5.

---

**Prompt A.5: Initial Answer Generation**

Answer the following question with just one entity.
Q: $Q$
A: {reasoning path} The answer is:

---

We use the substitute verification method to verify the correctness of the previous generated answer. Specifically, we first use the Prompt A.6 to identify the key condition within the question. By replacing the key condition with a specific token X, we create a masked question. We then append the sentence, "Suppose the answer is {previous generated answer}. What is the value of unknown variable X?" to the masked question to formulate the verification question, as shown in Prompt A.3.

---

**Prompt A.6: Key Condition Identification**

Given the question below, the task is to identify a set of entities within the question and then select the one that is most relevant to the problem-solving process.
$Q$

---

Using Prompt A.1 and Prompt A.5, we can obtain the answer for the verification question. By checking if the answer for the verification question and the key condition are equivalent, we can assess the correctness of the previous generated answer.

---

**Prompt A.7: Equivalence Check**

Determine the correctness of the proposition: If the answer to question {verification question} is {key condition}, then X could also be {answer for the verification question}

---

If the previous generated answer is deemed incorrect, we add it to the set of potentially incorrect answers; otherwise, we select it as the final answer. For incorrect answers, we can use the Prompt A.4 to correct them.

### A.3  Additional Experimental Results

**Can we just use the exact match method during the verification phase?**   Since verification questions can have multiple valid answers, directly checking if the LLM-generated response exactly matches the key condition might misclassify correct answers as incorrect. Consider the following example: Given an open-domain question *"Who wrote the treasure of the sierra madre?"*, we first prompt an LLM to generate an initial answer, e.g., *"B. Traven"*. Next, we identify a key condition in the question relevant to the problem-solving process, such as *"the treasure of the sierra madre"*. By masking the key condition, we create a verification question: *"Who wrote X? Suppose the answer is B. Traven. What is the value of unknown variable X?"*. Using the LLM to solve the verification question, we receive the response *"The Death Ship"*. If we directly check whether *"The Death Ship"*

matches *"the treasure of the sierra madre"*, we find they do not match, leading us to incorrectly judge the answer *"B. Traven"* as wrong. However, all books written by B. Traven are correct answers to the verification question. Thus, exact matching is insufficient for verification. Based on this observation, we propose proposition-based verification. Specifically, we construct an answer verification prompt: *"Determine the correctness of the proposition: If the answer to question "Who wrote X? Suppose the answer is B. Traven. What is the value of unknown variable X?" is "the treasure of the sierra madre", then X could also be "The Death Ship""*. We input this prompt into an LLM and receive a judgement about the proposition's correctness, e.g., *"The proposition is correct, since both works were written by the same author."*. This approach allows the LLM to properly analyze whether *"The Death Ship"* and *"the treasure of the sierra madre"* are both correct answers for the verification question, thus accurately determining the correctness of LLM-generated answers.

### A.4 Sample Predictions for Complex Reasoning Datasets

In this section, we provide sample predictions generated by the PROCO prompting method. For readability, we adjusted line breaks in the original text. As shown in Tables 8 to 15, PROCO accurately identifies and rectifies errors in LLM-generated answers without external feedback. Additionally, integrating RAG into PROCO significantly improves the accuracy and reliability of the answers.

| Question | Jean is two years older than Mark. Two years ago Mark was 5 years older than half Jan's age. If Jan is 30 how old is Jean? |
|---|---|
| **Gold Answer** | $\frac{30-2}{2} + 5 + 2 + 2 = 23$ |
| **Key Condition** | 5 |
| **Initial Answer** | Two years ago, Mark was 5 years older than half of Jan's age. Half of Jan's age two years ago would be 10 (30/2 - 2). Therefore, Mark's age two years ago would be 15 (10 + 5). We know that Jean is two years older than Mark. Therefore, Jean's age two years ago would be 17 (15 + 2). Jean's current age would be 19 (17 + 2). ✗ |
| **Verification Question** | Jean is two years older than Mark. Two years ago Mark was X years older than half Jan's age. If Jan is 30 how old is Jean? Suppose the answer is **19**. What is the value of unknown variable X? |
| **Answer to the Verification Question** | Two years ago Mark's age = Jean's age – 2 Half of Jan's age = 15 Mark's age two years ago = Jean's age – 2 = 15 – X Therefore, X = Jean's age – 15 – 2 = 19 – 15 – 2 = 2. Answer: **X = 2**. |
| **Verification Result** | Since 2 is not equal to the key condition, the answer 19 is likely **incorrect**. |
| **Question (with Hint)** | Jean is two years older than Mark. Two years ago Mark was 5 years older than half Jan's age. If Jan is 30 how old is Jean? **(The answer is likely not 19)** |
| **Corrected Answer** | Two years ago, Mark was 5 years older than half of Jan's age. Jan was 28. Half of Jan's age two years ago would be 14. Mark was 5 years older than 14 two years ago, so he was 19 two years ago. Jean is two years older than Mark, so Jean was 21 two years ago. Now, two years have passed since then, so Jean is 23 today. ✓ |
| **Verification Question** | Jean is two years older than Mark. Two years ago Mark was X years older than half Jan's age. If Jan is 30 how old is Jean? Suppose the answer is **23**. What is the value of unknown variable X? |
| **Answer to the Verification Question** | First, we're given that Jean is two years older than Mark. So if Mark is $M$ years old, Jean is $M + 2$ years old.
Two years ago, Mark was $M - 2$ years old and Jan's age was $30 - 2 = 28$ years.
We're also given that two years ago Mark was $X$ years older than half of Jan's age. So, $M - 2 = X + \frac{1}{2}(28)$.
Now, let's solve for $M$ using the information given. $$M - 2 = X + 14$$ $$M = X + 16$$ Now we know Mark's age in terms of $X$. Since Jean is two years older than Mark, Jean's age is $X + 18$.
If the answer is 23 for Jean's age, then: $$X + 18 = 23$$ $$X = 5$$ So, the value of the unknown variable **X is 5**. |
| **Verification Result** | Since 5 is equal to the key condition, the answer 23 is likely **correct**. |
| **Final Answer** | 23 ✓ |

Table 8: Example output obtained by PROCO on the GSM8K dataset.

| Question | In a class of 50 students, 28 participate in MATHCOUNTS, 21 participate in science club, and 6 students participate in neither. How many students participate in both MATHCOUNTS and science club? |
|---|---|
| **Gold Answer** | 5 |
| **Key Condition** | 28 |
| **Initial Answer** | Step 1: Find the total number of students who participate in either MATHCOUNTS or science club. Total students participating = 28 + 21 - 6 Total students participating = 43 Step 2: Calculate the number of students who participate in both MATHCOUNTS and science club. Number of students in both = 28 + 21 - 43 Number of students in both = 6 So, $\boxed{6}$ students participate in both MATHCOUNTS and science club. ✗ |
| **Verification Question** | First, we can calculate the number of students participating in either MATHCOUNTS or the science club or both: $50 - 6 = 44$. Next, we use the principle of inclusion and exclusion to set up our equation. The principle of inclusion and exclusion for two sets states: $$\|A \cup B\| = \|A\| + \|B\| - \|A \cap B\|$$ Here, $\|A \cup B\|$ represents the total number of students participating in either MATHCOUNTS or science club or both, which we have found to be $44$. $\|A\|$ represents the number of students in MATHCOUNTS ($X$), $\|B\|$ represents the number of students in science club (21), and $\|A \cap B\|$ represents the number of students in both clubs (6). Substituting these values into the equation, we get: $44 = X + 21 - 6$. Thus, the value of $X$ is $\boxed{29}$. |
| **Verification Result** | Since 29 is not equal to the key condition, the answer 6 is likely **incorrect**. |
| **Question (with Hint)** | In a class of 50 students, 28 participate in MATHCOUNTS, 21 participate in science club, and 6 students participate in neither. How many students participate in both MATHCOUNTS and science club? **(The answer is likely not 6)** |
| **Corrected Answer** | - Let $M$ represent the set of students participating in MATHCOUNTS. - Let $S$ represent the set of students participating in the science club. - Let $n$ represent the total number of students in the class, which is 50. First, we calculate the number of students participating in either MATHCOUNTS or the science club or both: $$\|M \cup S\| = n - \|N\| = 50 - 6 = 44$$ Using the principle of inclusion-exclusion for the union of two sets: $$\|M \cup S\| = \|M\| + \|S\| - \|M \cap S\|$$ Substitute the known values into the equation: $$44 = 28 + 21 - \|M \cap S\|$$ $$\|M \cap S\| = 5$$ Thus, the number of students who participate in both MATHCOUNTS and the science club is: $\boxed{5}$ ✓ |

Table 9: Example output obtained by PROCO on the MATH dataset.

| Question | A man spends Rs. 3500 per month and saves 12 1/2% of his income. His monthly income is ? Answer Choices: (A) Rs. 4400 (B) Rs. 4270 (C) Rs. 4000 (D) Rs. 3937.50 (E) None of these |
|---|---|
| Gold Answer | C |
| Key Condition | 3500 |
| Initial Answer | First, we need to calculate the amount saved each month. To do this, we can calculate 12 1/2% of Rs. 3500. 12 1/2% of Rs. 3500 = (12.5/100) * 3500 = 0.125 * 3500 = Rs. 437.50 So, the man saves Rs. 437.50 each month. To find his monthly income, we can use the formula: Income = Expenditure + Savings Income = Rs. 3500 + Rs. 437.50 Income = Rs. 3937.50 Therefore, his monthly income is Rs. 3937.50, which corresponds to answer choice (D). ✗ |
| Verification Question | A man spends Rs. X per month and saves 12 1/2% of his income. His monthly income is ? Answer Choices: (A) Rs. 4400 (B) Rs. 4270 (C) Rs. 4000 (D) Rs. 3937.50 (E) None of these Suppose the answer is (D) Rs. 3937.50. What is the value of unknown variable X? |
| Answer to the Verification Question | 1. Convert the percentage saved into a decimal: 12.5% can be expressed as: $$12.5\% = \frac{12.5}{100} = 0.125$$ 2. Determine the amount saved: Since he saves 12.5% of his income, the amount saved each month is: $$0.125 \times 3937.50 = 492.1875$$ 3. Calculate the amount spent (X): The amount he spends is the remaining part of his income after saving. Therefore, the amount spent $X$ is: $$X = \text{Income} - \text{Amount saved} = 3445.3125$$ The value of the unknown variable $X$ (the amount spent per month) is: $\boxed{3445.3125}$ |
| Verification Result | Since 3445.3125 is not equal to the key condition, the answer (D) Rs. 3937.50 is likely **incorrect**. |
| Question (with Hint) | A man spends Rs. 3500 per month and saves 12 1/2% of his income. His monthly income is ? Answer Choices: (A) Rs. 4400 (B) Rs. 4270 (C) Rs. 4000 (D) Rs. 3937.50 (E) None of these **(The answer is likely not (D) Rs. 3937.50)** |
| Corrected Answer | Let $I$ be the man's monthly income. Since he saves 12.5% of his income, he spends the remaining percentage of his income. The percentage of income spent is: $$100\% - 12.5\% = 87.5\%$$ $$\text{Income spent} = 0.875 \times I$$ We know that the amount spent is Rs. 3500. Thus, we set up the equation: $$0.875 \times I = 3500$$ To find $I$, we solve the equation for $I$: $$I = \frac{3500}{0.875} = 4000$$ The man's monthly income is Rs. 4000. The correct answer is: (C) Rs. 4000 ✓ |

Table 10: Example output obtained by PROCO on the AQuA dataset.

| Question | Right to property according to the constitution of india is a? |
|---|---|
| **Gold Answer** | constitutional right |
| **Key Condition** | property |
| **Initial Answer** | The right to property in the Constitution of India has undergone significant changes over the years. Prior to the 44th Amendment Act in 1978, the right to property was considered a fundamental right under Part III of the Constitution. However, this amendment removed the right to property from the list of Fundamental Rights and converted it into a simple legal right (https://byjus.com/question-answer/consider-the-following-statements-regarding-right-to-property-1-right-to-property-is-legal-right/) ✗ |
| **Verification Question** | Right to X according to the constitution of india is a? Suppose the answer is simple legal right. What is the value of unknown variable X? |
| **Answer to the Verification Question** | the right to legal representation. |
| **Verification Result** | Since the right to legal representation and the property are not equivalent, the answer simple legal right is likely **incorrect**. |
| **Question (with Hint)** | Right to property according to the constitution of india is a? **(The answer is likely not simple legal right)** |
| **Corrected Answer** | The right to property in the Constitution of India has evolved from being a fundamental right to a legal and constitutional right, with provisions for legal protection and acquisition for public welfare. (https://testbook.com/question-answer/what-is-the-position-of-the-right-to-property-in-i–616294b1fb5e57239764c589) ✓ |
| **Verification Question** | Right to X according to the constitution of india is a? Suppose the answer is constitutional right. What is the value of unknown variable X? |
| **Answer to the Verification Question** | In the context of the Constitution of India, the unknown variable X refers to a specific right that is classified as a "constitutional right." Given this information, one possible value for X is "property." To elaborate, the right to property was originally a fundamental right under Article 31 of the Indian Constitution. However, it was later amended by the 44th Amendment Act of 1978, and the right to property was removed from the list of fundamental rights and reclassified as a constitutional right under Article 300A. Therefore, the value of the unknown variable X is **"property."** |
| **Verification Result** | Since the property and the property are equivalent, the answer constitutional right is likely **correct**. |
| **Final Answer** | constitutional right ✓ |

Table 11: Example output obtained by RAG + PROCO on the NQ dataset.

| Question | Who was the defending champion when Martina Navratilova first won Wimbledon singles? |
|---|---|
| **Gold Answer** | Virginia Wade |
| **Key Condition** | Martina Navratilova |
| **Initial Answer** | When Martina Navratilova first won the Wimbledon singles title, the defending champion was Chris Evert (https://en.wikipedia.org/wiki/1977_Wimbledon_Championships). ✗ |
| **Verification Question** | Who was the defending champion when X first won Wimbledon singles? Suppose the answer is Chris Evert. What is the value of unknown variable X? |
| **Answer to the Verification Question** | Given that Chris Evert is the answer, X must be the name of a person who won Wimbledon singles. Conclusion: X refers to a person who won Wimbledon singles, but without more information, the specific name cannot be determined. |
| **Verification Result** | Since unknown and Martina Navratilova are not equivalent, the answer Chris Evert is likely **incorrect**. |
| **Question (with Hint)** | Who was the defending champion when Martina Navratilova first won Wimbledon singles? **(The answer is likely not Chris Evert)** |
| **Corrected Answer** | The defending champion when Martina Navratilova first won the Wimbledon singles title in 1978 was Virginia Wade (https://en.wikipedia.org/wiki/1978_Wimbledon_Championships_–_Women's_singles). ✓ |
| **Verification Question** | Who was the defending champion when X first won Wimbledon singles? Suppose the answer is Virginia Wade. What is the value of unknown variable X? |
| **Answer to the Verification Question** | Martina Navratilova won her first Wimbledon singles title in 1978. Virginia Wade was the Wimbledon singles champion in 1977. So, if we are to match the scenario where X first won Wimbledon when Virginia Wade was the defending champion: In 1978, the defending champion was Virginia Wade (from her 1977 win). Martina Navratilova won Wimbledon singles in 1978. Therefore, ( X ) is indeed Martina Navratilova. Thus, the value of the unknown variable ( X ) is **Martina Navratilova**. |
| **Verification Result** | Since Martina Navratilova and Martina Navratilova are equivalent, the answer Virginia Wade is likely **correct**. |
| **Final Answer** | Virginia Wade ✓ |

Table 12: Example output obtained by RAG + PROCO on the TriviaQA dataset.

| | |
|---|---|
| **Question** | What highschool did harper lee go to? |
| **Gold Answer** | Monroe County High School |
| **Key Condition** | highschool |
| **Initial Answer** | Harper Lee attended several educational institutions during her academic journey. According to the information available, she attended Huntingdon College from 1944 to 1945. After that, she studied law at the University of Alabama from 1945 to 1949. Additionally, she spent a year at Oxford University (`https://www.tokillamockingbird.com/harper-lee`). Based on the available information, it can be concluded that Harper Lee did not attend high school. She pursued higher education after completing her grade school education. Conclusion Harper Lee did not attend high school. ✗ |
| **Verification Question** | What X did harper lee go to? Suppose the answer is Harper Lee did not attend high school. What is the value of unknown variable X? |
| **Answer to the Verification Question** | If the statement "Harper Lee did not attend high school" is true, then the value of the unknown variable X, representing the high school she attended, would be **"none."** This means Harper Lee did not go to any high school. |
| **Verification Result** | Since none and highschool are not equivalent, the answer Harper Lee did not attend high school is likely **incorrect**. |
| **Question (with Hint)** | What highschool did harper lee go to? (**The answer is likely not Harper Lee did not attend high school**) |
| **Corrected Answer** | Nelle Harper Lee, the acclaimed author of "To Kill a Mockingbird," attended monroe county high school in her hometown of Monroeville, Alabama. She was born on April 28, 1926, and grew up in a rural southern Alabama town, where she attended both elementary school and high school. Her father, Amasa Coleman Lee, was an attorney who served in the state legislature in Alabama, and her childhood experiences were similar to those of her young narrator in "To Kill a Mockingbird," Scout Finch (`https://www.chipublib.org/harper-lee-biography/`) ✓ |
| **Verification Question** | What X did harper lee go to? Suppose the answer is monroe county high school. What is the value of unknown variable X? |
| **Answer to the Verification Question** | If the answer is "Monroe County High School," the value of the unknown variable X is "**high school**." |
| **Verification Result** | Since high school and highschool are equivalent, the answer monroe county high school is likely **correct**. |
| **Final Answer** | monroe county high school ✓ |

Table 13: Example output obtained by RAG + PROCO on the WebQ dataset.

| | |
|---|---|
| **Question** | What could go on top of wood? Answer Choices: (A) lumberyard, (B) synagogue, (C) floor, (D) carpet, (E) hardware store |
| **Gold Answer** | D |
| **Key Condition** | wood |
| **Initial Answer** | To determine what could go on top of wood, let's analyze the answer choices provided: (A) Lumberyard: A lumberyard is a facility where wood and lumber are stored and sold. It is not something that goes on top of wood. (B) Synagogue: A synagogue is a place of worship for Jewish people. It is not something that goes on top of wood. (C) Floor: Yes, a floor can go on top of wood. Wood is commonly used as a material for flooring, providing a natural and warm aesthetic. Hardwood floors, for example, are made of solid wood and are installed on top of a subfloor. (D) Carpet: Yes, carpet can go on top of wood. Carpet is often installed on top of a wooden subfloor, providing a soft and comfortable surface. (E) Hardware store: A hardware store is a retail store that sells tools, building materials, and other hardware items. It is not something that goes on top of wood. Therefore, the correct answer is ==(C) floor==. ✗ |
| **Verification Question** | What could go on top of X? Suppose the answer is floor. What is the value of unknown variable X? |
| **Answer to the Verification Question** | The question asks what could go on top of X, and the answer is "floor." Based on this, we can infer that X refers to a "ceiling." The ceiling is the upper surface of a room and is what goes on top of the floor. Therefore, X refers to **ceiling**. |
| **Verification Result** | Since wood and ceiling are not equivalent, the answer (C) floor is likely **incorrect**. |
| **Question (with Hint)** | What could go on top of wood? Answer Choices: (A) lumberyard, (B) synagogue, (C) floor, (D) carpet, (E) hardware store **(The answer is likely not (C) floor)** |
| **Corrected Answer** | The answer is likely not in ['C']. Based on the provided search results, the item that could go on top of wood is ==(D) carpet==. ✓ |
| **Verification Question** | What could go on top of X? Suppose the answer is carpet. What is the value of unknown variable X? |
| **Answer to the Verification Question** | If the answer to "What could go on top of X?" is "carpet," then X is likely the floor or flooring. Therefore, the value of the unknown variable X is **floor**. |
| **Verification Result** | Yes, the proposition is correct. If the answer to the question "What could go on top of X? Suppose the answer is carpet" is "wood", then X could also be "floor". 
Here's the reasoning: 1. The question asks what could go on top of X, and the given answer is "carpet". 2. This implies that "carpet" can be placed on top of X. 3. The follow-up statement tells us that the answer to this is "wood". Therefore, wood is X. 4. Carpet is typically placed on top of a floor surface. 5. Wood is a common type of flooring. 
Thus, if wood is the answer to X, and considering the context of flooring, X being "floor" is a reasonable interpretation because a carpet is often placed on a wood floor. 
Therefore, the unknown variable X, which in this context is the surface on which carpet can be placed, could be "floor". 
Since floor and wood are equivalent, the answer (D) carpet is likely **correct**. |
| **Final Answer** | D ✓ |

Table 14: Example output obtained by PROCO on the CSQA dataset.

| Question | The trophy given to the winner of the college football game between University of Idaho Vandals and University of Montana Grizzlies is derived from which European culture? |
|---|---|
| **Gold Answer** | German |
| **Key Condition** | European culture |
| **Initial Answer** | The University of Idaho Vandals and the University of Montana Grizzlies are both located in the United States. However, the trophy's origin may be derived from a European culture due to the historical influence of European traditions on American sports and collegiate traditions. ✗ |
| **Verification Question** | The trophy given to the winner of the college football game between University of Idaho Vandals and University of Montana Grizzlies is derived from which X ? Suppose the answer is European culture. What is the value of unknown variable X? |
| **Answer to the Verification Question** | Determine which aspect of European culture matches the question about the trophy. Given that the trophy is derived from European culture, X must be a specific aspect or tradition within European culture. X is **"Medieval armor and weaponry"**. |
| **Verification Result** | Since Medieval armor and weaponry and European culture are not equivalent, the answer European culture is likely **incorrect**. |
| **Question (with Hint)** | The trophy given to the winner of the college football game between University of Idaho Vandals and University of Montana Grizzlies is derived from which European culture? **(The answer is likely not European culture)** |
| **Corrected Answer** | First, it's important to note that the University of Idaho Vandals and the University of Montana Grizzlies are both located in the United States, so the trophy is likely derived from American culture rather than European culture. The trophy for this game is called the Little Brown Stein, and it is a tradition that dates back to 1938. The name "Stein" suggests a German influence, as "stein" is a German word for "stone" or "mug." Therefore, the trophy is likely derived from German culture. ✓ |
| **Verification Question** | The trophy given to the winner of the college football game between University of Idaho Vandals and University of Montana Grizzlies is derived from which X ? Suppose the answer is German culture. What is the value of unknown variable X? |
| **Answer to the Verification Question** | If the trophy given to the winner of the college football game between the University of Idaho Vandals and the University of Montana Grizzlies is derived from German culture, the unknown variable X could be "**German traditions**." |
| **Verification Result** | Given the information provided, if the answer to the question is indeed "European culture", then X could indeed be "German traditions". This is because German traditions could fall under the umbrella of European culture. German traditions are a subset of European culture, so if the answer is broadened from "European culture" to a more specific subset like "German traditions", it would still be consistent with the broader category. Therefore, the proposition appears to be correct. 
 Since European culture and German traditions are equivalent, the answer German culture is likely **correct**. |
| **Final Answer** | German culture ✓ |

Table 15: Example output obtained by PROCO on the HotpotQA dataset.

