# OpenReview forum: "Large Language Models Can Self-Correct with Minimal Effort"
_ICML.cc/2024/Workshop/AI4MATH — ICML 2024 Workshop AI4MATH Poster_

### Official Review · Reviewer_9jQj · 2024-06-12

**Rating:** 8
**Confidence:** 4

**Summary:**

This paper proposes a method ProCo (Pro) to verify the correctness of LLM's output for reasoning tasks such as mathematical reasoning and open domain question answering. For an output answer given by an LLM for a reasoning question, the approach constructs a backward reasoning task and let the LLM to solve this new task. Then the consistency between the original question/answer and the new question/answer is checked and this serves as a way for correctness verification. The new task is constructed as follows. First a key condition of the question is identified and a key word in the condition is masked. A sentence with the output answer is appended and a question on the masked part is posed. If the consistency is not satisfied, then this answer is added to a list of unpromising answers and a hint is appended to the prompt stating "these answers are not promising" etc. ProCo is evaluated on benchmarks of math reasoning, open-domain question answering and commonsense reasoning and has consistent improvement against baseline methods such as CoT and Self-Correct.

**Questions:**

1. The proposed method seems a combination of MetaMath [1] for masking key conditions and progressive-hint prompting (PHP) [2] for giving progressive hint. Are you familar with these papers? Is the approach inspired by them? In any case, it would be good to add these papers to related work.

[1] Yu L, Jiang W, Shi H, et al. MetaMath: Bootstrap Your Own Mathematical Questions for Large Language Models.
[2] Zheng C, Liu Z, Xie E, Li Z, Li Y. Progressive-Hint Prompting Improves Reasoning in Large Language Models.

**Reasons To Accept:**

1. The approach is well-motivated and the proposed pipeline makes sense.
2. The paper is written in a clear manner and easy to follow
3. The experimental results are extensive and the improvements are consistent and significant
4. The idea seems to creatively combine existing methods and does not require a formal system for verification. This is a bit surprising
5. Extensive case study is provided.

**Reasons To Reject:**

1. It is not analyzed how many rounds the progressive hint process will take. This distribution is important for analyzing the required computational resource.

---

### Official Review · Reviewer_QCRi · 2024-06-12

**Rating:** 8
**Confidence:** 4

**Summary:**

Can language models improve their answers using their own feedback signals? It is this question that the authors target. The authors propose a clever, counterfactual-esque feedback method whereby a model's answer is revealed back to itself, reformulated by asking the model to assess whether the answer makes sense relative to a key unknown in the question. The authors dub their method ProCo and find that it substantially improves language model capabilities across a range of questions.

**Questions:**

I included several questions above, but my biggest question is what tasks the authors think ProCo would not be well-suited for. This is important to discuss.

**Reasons To Accept:**

ProCo is a clever technique that promises broad impact, not just in mathematics but other commonsense reasoning domains, as the authors discuss. The authors conduct a wonderfully extensive set of experiments and analyses, including a range of ablations and comparisons across a suite of datasets, benchmarks, and base models. Well done!

I am especially excited by Figure 4 -- reducing the rate of iterative-feedback taking questions that are correctly answered to incorrect is important.

**Reasons To Reject:**

This paper should not be rejected, in my opinion. However, there are a few things that could be improved: (i) it would be good for the authors, were possible, to add error bars; (ii) I found some of the tables visually cluttered -- the key comparison is ProCo against other methods -- consider bolding within subsets -- e.g., what's better in the GenRead comparisons (baseline or ProCo) -- rather than bolding over the whole table; (iii) perhaps most importantly, I feel like a discussion of limitations is missing. What kinds of tasks would ProCo fail at? It seems being able to identify the "key concept" could be a key bottleneck?

---

### Official Review · Reviewer_Dx9t · 2024-06-13

**Rating:** 6
**Confidence:** 1

**Summary:**

propose an iterative verify-then-correct framework to progressively identify and correct (probably) false responses

**Questions:**

1. No

**Reasons To Accept:**

1. Compared with Self-Correct，propose the concept of key condition and verification

**Reasons To Reject:**

1. Not very innovative

---

### Meta-Review · Area_Chair_As31 · 2024-06-13

**Recommendation:** Accept (Poster)
**Confidence:** 4

**Metareview:**

**Summary:**
The paper presents ProCo, a feedback method for language models to improve their responses by using their feedback signals. ProCo works by taking the model's initial answer and reformulating it into a new task where the model assesses the consistency of the answer relative to a key unknown in the question. The authors demonstrate that ProCo significantly enhances language model performance across various reasoning tasks, including mathematical reasoning and open-domain question answering.

**Strengths:**

1. ProCo is a clever and promising technique that shows potential for broad applicability beyond mathematics to other commonsense reasoning domains.
2. The paper includes extensive experiments, ablations, and comparisons across multiple datasets, benchmarks, and base models, providing a robust validation of ProCo's effectiveness.
3. The paper is well-written, with clear explanations of the method and its implications. The inclusion of detailed case studies and extensive experimental results strengthens the presentation.
4. The experimental results indicate consistent and significant improvements over baseline methods such as CoT and Self-Correct.
5. The approach creatively combines elements from existing methods without requiring a formal system for verification, which is an innovative aspect.

**Areas for Improvement:**

1. Adding error bars to the experimental results would enhance the clarity and reliability of the findings.
2. Some tables are visually cluttered. Highlighting key comparisons within subsets (e.g., ProCo vs. baselines in specific scenarios) rather than bolding across the entire table could improve readability.
3.  A discussion on the limitations of ProCo is missing. It would be beneficial to understand what types of tasks ProCo may not be well-suited for, particularly regarding identifying key concepts.
4. There is a lack of analysis of the computational resources required, especially regarding the distribution of rounds in the progressive hint process. This information is crucial for evaluating ProCo's practical applicability.

---

### Decision · Program_Chairs · 2024-06-13

Accept (Poster)